# Intrusive social support among Black and White individuals with type 2 diabetes: A "Control issue" or a sign of "Concern and love"?

**Jeanean B. Naqvi**[1]*, **Rachael S. Liu**[1], **Vicki S. Helgeson**[1], **Megan E. Hamm**[2]

**1** Department of Psychology, Carnegie Mellon University, Pittsburgh, Pennsylvania, United States of America, **2** Department of General Internal Medicine, University of Pittsburgh, Pittsburgh, Pennsylvania, United States of America

* jnaqvi@health.ucsd.edu

## Abstract

Family members and friends play an important supportive role in the management of chronic illnesses like diabetes, which often require substantial lifestyle changes. Some studies suggest that there may be racial differences in the kinds of support people receive, though little research has examined this idea within a chronic illness context. The current research takes a qualitative approach to examining similarities and differences between Black and White individuals with type 2 diabetes in the dimensions of support received from their family members, with a particular focus on better understanding more intrusive forms of support, such as unsolicited and overprotective support. Semi-structured interviews were conducted (N = 32) to characterize differences in support received by Black and White individuals with type 2 diabetes. The results of the thematic analysis suggested that unsolicited and overprotective support were not universally perceived to be negative, as previous work on White populations seemed to suggest. Rather, if the support provided was perceived as inhibiting autonomy, it was generally undesired by participants from both racial groups—however, for Black participants, knowing that the support was provided out of love could make it more acceptable. The analysis also revealed several underexplored dimensions of received support, including the directiveness of support and the tone used to deliver support. The current study provides an initial step towards grounding social support theory in the experiences of marginalized populations and will inform further development of a culturally sensitive measure of social support for individuals with chronic illness.

## Introduction

About 34.2 million people in the United States have diabetes [1]—an illness that requires a complicated self-management regimen of adherence to medication, blood glucose checking, and doctor's visits, as well as changes to diet and physical activity [2]. Non-Hispanic Black individuals suffer a disproportionate burden of diabetes prevalence [1], diabetes-related

**Data Availability Statement:** Data cannot be shared publicly due to ethical restrictions. The interview transcripts contain personally identifiable information that cannot be redacted without

removing content relevant to the research. However, anonymized data excerpts can be accessed upon reasonable request by contacting the Carnegie Mellon University Institutional Review Board (irb-review@andrew.cmu.edu).

**Funding:** JN received funding from the National Institute on Minority Health and Health Disparities (https://www.nimhd.nih.gov/) F31MD015922, the National Heart, Lung, and Blood Institute (https://www.nhlbi.nih.gov/) T32HL079891, and the Society for Personality and Social Psychology (https://spsp.org/) Jenessa Shapiro Graduate Research Award. The funders had no role in study design, data collection and analysis, decision to publish, or preparation of the manuscript.

**Competing interests:** The authors have declared that no competing interests exist.

complications like retinopathy and lower limb amputation [3], and diabetes-related comorbidities like cardiovascular disease or chronic kidney disease [4] compared to non-Hispanic White individuals. However, receiving social support from close others has shown vast benefits for both non-Hispanic Black and non-Hispanic White individuals with diabetes, including better self-management behaviors, reduced diabetes distress and lower HbA1c [5]. Despite these benefits, studies have rarely explored whether there may be race differences in the characteristics or dimensions of social support received by Black and White individuals. To address this gap in the literature, the current study used an inductive approach to compare the dimensions of social support received from family by Black and White individuals with type 2 diabetes, focusing particularly on more intrusive forms of support like unsolicited support (support that is unasked for) and overprotective support (support that underestimates the person's capabilities).

## Race differences in support

There is reason to believe that the expectations around social support might differ between Black and White individuals. Because the discrimination Black individuals face on a daily basis threatens their economic success [6], their well-being [7], and their existence [7, 8], these hardships could affect the nature of their support interactions. For example, African Americans actively exchange support within family networks, such as sharing childcare responsibilities, providing financial help, and assisting with transportation [9], which may have initially developed as a strategy for survival in low socioeconomic communities [10]. Over time, certain kinds of support that were originally enacted as a way to survive may have become more culturally normative. Thus, Black individuals may have developed expectations for receiving different kinds of support than White individuals.

Traditionally, social support behaviors have often been characterized in the literature based on type—for example, family members may offer tangible assistance and provision of resources (instrumental support), reassurance and emotional concern (emotional support), or advice and suggestions (informational support) [11]. Only a few studies have investigated whether there are differences between Black and White individuals in these types of social support, and results are mixed. Some studies have found higher provision of instrumental support among Black individuals than White individuals [12, 13]. Regarding emotional support, one study found higher perceived emotional support among African Americans than White individuals [14], another found higher perceived emotional support among White individuals than Black individuals [15], and yet another found no group difference [16]. Thus, prior research does not indicate a clear pattern of findings.

## Race differences in intrusive support

Based on the previous section, the literature examining Black/White differences in levels of social support is sparse and only focuses on certain types of social support, like instrumental support and emotional support. However, there are likely other dimensions of support that are salient to individuals and important for health. One support dimension that may be interesting to consider is "intrusiveness," a form of support that has been studied at length within the developmental literature and refers to parental attempts at support that interfere with or restrict a child's autonomy [17, 18]. Researchers often classify multiple behaviors as intrusive [19], including unsolicited support—support that is unasked for—and overprotective support —support that appears to underestimate the person's capabilities. However, previous researchers have theorized that intrusiveness may have different meanings in different cultural groups and may be more acceptable or adaptive in certain contexts [19]. Given the social context in

which Black youth are more likely to be suspended from school [20], be arrested [21], and experience police brutality [22] than White youth, Black parents may be providing unsolicited and overprotective support in an effort to safeguard children from societal harm. Indeed, several previous studies with children demonstrate that Black children experience more maternal intrusiveness than White children [17, 23–25].

Though a parent-child relationship is certainly distinct from an adult's relationship to their family, Black adults face some of the same concerns as Black youth, including a higher likelihood of arrest and experience of police brutality than White individuals [8]. Thus, Black families may exercise similar kinds of "intrusive" support towards their kin to protect them from societal harm. However, research outside of the developmental context has not considered intrusive behaviors to be "support," examining them instead as part of "negative interactions" or "unsupportive behaviors." The results of these studies are mixed as well: two studies found less frequent negative family interactions among African Americans than White individuals [14, 26], another study found less frequent negative interactions among White individuals than Black individuals [12], and other studies found no differences in negative interactions [15, 16] or unsupportive behaviors [27] between Black and White individuals. One possible explanation for the mixed findings is that the measures of "negative" or "unsupportive" behaviors mix together intrusive behaviors that may be intended to help, like criticizing an individual, and behaviors that are likely not intended to help, like letting the individual down. A few other studies have specifically examined overprotectiveness within the context of diabetes and found direct and indirect links between overprotectiveness and increased diabetes-specific distress, but these studies were conducted with predominantly European or European American samples [28–30]. Given these gaps in the literature, it is important to more closely examine the experience of intrusive behaviors among Black individuals as well, particularly those coping with diabetes.

## The present research

Research on social support has focused primarily on White individuals, using theories and measures based on White samples. The research that examines race differences in support focuses primarily on type of support (e.g., instrumental, emotional) as the most salient support dimension. Using these measures and theories in non-White groups reveals an assumption of racial/ethnic similarity [31]—that social support is likely experienced similarly across racial/ethnic groups—without first being empirically tested. However, there may be other support dimensions (e.g., intrusiveness) experienced differently by non-White racial/ethnic groups and some unexplored support dimensions that are more salient to non-White racial/ethnic groups. To explore this issue, we utilized qualitative interviews to inductively examine similarities and differences between Black and White individuals with type 2 diabetes in the dimensions of support received from their family. Because our primary interest was in intrusive forms of support, some of our main interview questions were focused on understanding what kind of support behaviors were perceived to be unsolicited or overprotective. However, the inductive nature of this study allowed us to uncover additional support dimensions that were salient to participants. We were also able to explore racial differences in other themes related to support, including participants' feelings towards various dimensions of received support and beliefs about the underlying reasons that support was provided.

## Method

### Participants

The final sample for our analysis included 32 participants with type 2 diabetes with an equal distribution of participants by gender (50% female) and race (50% Black). Participants were 59

**Table 1. Demographics.**

|  | Overall (*N* = 32) | White (*n* = 16) | Black (*n* = 16) |
|---|---|---|---|
| Gender; *n* (% female) | 16 (50%) | 8 (50%) | 8 (50%) |
| Age (years); mean (SD) | 59.06 (1.04) | 63.25 (6.36) | 54.88 (11.42) |
| Years since diagnosis; mean (SD) | 10.19 (6.49) | 9.81 (4.98) | 10.56 (7.87) |
| On insulin; *n* (%) | 9 (28%) | 4 (25%) | 5 (31%) |
| Marital status; *n* (% married/in a relationship) | 24 (75%) | 13 (81%) | 11 (69%) |
| Employment; *n* (% currently working) |  |  |  |
| Employed | 11 (34%) | 8 (50%) | 3 (19%) |
| Not employed | 8 (25%) | 1 (6%) | 7 (44%) |
| Retired | 11 (34%) | 6 (38%) | 5 (31%) |
| Household income; *n* (%): |  |  |  |
| Less than $20,000 | 5 (16%) | 0 (0%) | 5 (31%) |
| $20,000 - $49,999 | 11 (34%) | 4 (25%) | 7 (44%) |
| $50,000 - $149,999 | 10 (31%) | 9 (56%) | 1 (6%) |
| $150,000 and above | 1 (3%) | 1 (6%) | 0 (0%) |
| Decline to answer | 5 (16%) | 2 (13%) | 3 (19%) |
| Education level; *n* (%): |  |  |  |
| High school graduate/GED | 3 (9%) | 0 (0%) | 3 (19%) |
| Some college | 7 (22%) | 1 (6%) | 6 (38%) |
| College graduate | 22 (69%) | 15 (94%) | 7 (44%) |
| Number of family members in the home; mean (SD) | 1.34 (1.10) | 1.31 (1.01) | 1.38 (1.20) |
| Number of family members seen regularly; mean (SD) | 5.81 (4.34) | 6.81 (4.04) | 4.81 (4.52) |

years of age on average (*SD* = 10.04), and the majority were college graduates (69%). The majority were married or in a relationship (75%). About one-third (34%) were currently employed, and about one-third (34%) were retired. Full demographic information is included in Table 1.

## Recruitment

Purposive sampling was conducted via an ongoing type 2 diabetes study [32] and through the University of Pittsburgh Pitt+Me® Research Participant Registry (see S1 Fig for recruitment diagram). Participants recruited from the ongoing type 2 diabetes study were contacted via phone to determine their interest in the study, whereas participants recruited from the Pitt+Me® Research Participant Registry viewed a description of the study on the Pitt+Me® website and indicated their eligibility and interest in participating via the website. Individuals were screened over the phone and needed to identify as either Black/African American or White, self-report that they were diagnosed with type 2 diabetes, and have at least one family member or person that they considered to be family who supported them with diabetes. Given that individuals with diabetes are vulnerable to COVID-19 [33], participants also needed to have an internet-connected computer or smartphone to participate in the interview remotely via video call or phone call. One individual was included despite having borderline diabetes rather than being officially diagnosed with type 2 diabetes.

Thematic saturation, a common criterion for determining a qualitative sample size, is typically reached at 12 to 16 interviews [34]. Thus, our goal was to interview at least 16 Black and 16 White participants. We also aimed to have a balanced representation of gender within each racial group.

## Data collection

The study was approved by the Carnegie Mellon University Institutional Review Board, and all study participants provided written consent using an online consent form prior to participating in the study. Participants engaged in a 1-hour, semi-structured interview conducted by members of the study team (J.N. and R.L.), who were trained in interview techniques by an experienced qualitative investigator who was also part of the study team (M.H.). At the beginning of the interview, the interviewer reiterated the purpose of the study and asked that the participant be a private area of their home and out of earshot from other people living in their home for confidentiality reasons. Main interview topics centered around diabetes management and support interactions with their family members regarding diabetes (see Supporting Information). To help participants focus on and recall specific support interactions they remembered having with their close family, the interviewer presented the participant with a visual depiction of Kahn and Antonucci's [35] convoy model. Traditionally viewed as a heuristic framework for understanding social relationships, some researchers have suggested it may be a useful cognitive aid for discussing relationships within the context of qualitative interviews [36]. The model depicted four concentric circles: the word "YOU" was in the innermost circle, and each subsequent circle was labeled "Inner Circle", "Middle Circle", and "Outer Circle." The interviewer defined each of these circles in the following way: the inner circle represented "people who are like family to you that you feel so close to that it's hard to imagine life without them"; the middle circle represented "the next closest group of people that are like family to you"; and the outer circle represented "the group of people who are least close to you but are still part of your family network." Presenting the model in this way allowed us to ask questions about support interactions with anyone they considered to be family, not just with individuals related to the participant by blood. We saw this as an important distinction given that fictive kin have been shown to be an integral part of Black families [37].

Subsequent sections of the interview were primarily focused on interactions with the participant's inner circle. Participants were asked questions about behaviors in which people in their inner circle had engaged in to support the participant with diabetes without the participant asking them (unsolicited support), as well as behaviors in which people in their inner circle had tried to do that the participant could do themselves or behaviors that restrict the participant's ability to do certain things (overprotective support). Interview probes included questions about how these support interactions made them feel, as well as what they saw as the reasons behind their inner circle members' behaviors. Participants were later asked a few questions about the support interactions they had with people in their middle circle and outer circle. However, most participants either reported minimal support interactions from these circles or that the support interactions were largely the same as those experienced with their inner circle. As such, we focus on the patterns of results from the inner circle support interactions that participants discussed.

The interviews were conducted from January to March 2021. Following each interview, background information was collected by administering a demographic questionnaire, and participants were debriefed and told that a summary of the findings would be provided after study completion. Study team members who conducted the interviews (J.N. and R.L.) completed a field note immediately after each interview and met on a bi-weekly basis to discuss their notes. Interview sessions were audio-recorded and transcribed verbatim by a research assistant with identifying information redacted from the transcript. Participants received $25 in compensation.

## Data analysis

After data collection was complete, a combined inductive and deductive coding approach was used to conduct thematic analysis [38]. The primary and secondary coders (J.N. and R.L.) developed a preliminary codebook based on 12 (38%) of the transcripts. The codebook consisted of the main topics touched upon by participants (e.g., diabetes diagnosis, feelings about diabetes, diabetes management activities, inner circle support behaviors), as well as a set of codes inferred from the participants' language (e.g., support tone). Then, the coders generated more fine-grained codes, combined higher order codes as needed, and developed definitions for each code. Deductive codes were added based on prior categorizations in the literature (e.g., support behaviors were coded as instrumental, emotional, or informational support; see Supplemental Information for the final coding system). Next, both the primary and secondary coders independently coded a random subset of 8 transcripts (25%) and resolved any discrepancies using consensus adjudication. Mean percentage of agreement was 80%. Once the primary and secondary coders' reliability was established, the primary coder proceeded to code the 24 remaining transcripts. Both coders used MAXQDA Version 20.4.0 to assist with data management.

Once the interviews were coded, the primary coder identified broad themes present throughout the interviews, specifically looking for similarities and differences in thematic patterns by race. These themes were then discussed with a member of the study team (M.H.) and another investigator (V.H.) who had not participated in coding but was involved in the development of the interview guide. The primary coder then identified and defined subthemes that were refined through further discussion with M.H. and V.H. Dependability and confirmability were addressed by documenting each step of the analytic process via an audit trail.

The results are organized into five main themes. Direct quotes from participants are accompanied by basic demographic information, and all names have been replaced by pseudonyms.

## Positionality

As with all research, it is important to understand the positionality of the investigators involved in this project [39]. Of the study team members who conducted the interviews and coded the data, one was an undergraduate research assistant who identified as an Asian American cisgender woman and the other was a graduate student who identified as a mixed-race cisgender woman (Asian American and White American). Themes were then discussed with two members of the study team who were White cisgender women. Thus, the study team did not have the advantages of an insider position when investigating the experiences of Black participants in this study, such as having more insight into subtext, non-verbal cues, or a priori knowledge of the culture, nor were they able to establish trust with participants on the basis of the shared experience of race. However, there are advantages of having an outsider position, such as being able to bring an external perspective to understanding Black participant experiences, and an ability to ask direct questions about cultural phenomena that may have been obvious to an insider. Neither set of advantages necessarily outweighs the other but should be kept in mind when considering the results of this study.

## Results

Interview questions centered around unsolicited and overprotective support, our primary dimensions of interest. However, the coding process revealed two novel dimensions of support: *support directiveness*, which refers to the extent to which participants discussed being told what to do versus being offered suggestions, and *support tone*, which refers to the tone that participants perceived close others as using when providing support. Participants'

discussions of the support they received centered around five main themes: 1) across participants from both racial groups, unsolicited emotional support was perceived positively, but reactions towards unsolicited instrumental and informational support were variable; 2) when it came to overprotective support, participants from both racial groups generally found close others doing things for them to be helpful, whereas participants' feelings about restrictions varied by race; 3) in terms of support directiveness, participants from both racial groups generally expressed negative feelings when close others told them what to do, even if they found it helpful; 4) if support was perceived as inhibiting autonomy, it was generally undesired—but for Black participants, knowing that the support was provided out of love and care could make it more acceptable; and 5) participants from both racial groups seemed to describe the tone that close others used to provide support, and this tone allowed participants to infer the reasons for the support being provided and may impact feelings towards support.

### Theme 1: Across participants from both racial groups, unsolicited emotional support was perceived positively, but reactions towards unsolicited instrumental and informational support were variable

Both Black and White participants discussed a wide variety of support interactions, each of which could be described by several different support dimensions. Most pertinent to our primary research question, these support interactions varied regarding whether they were solicited. Many participants described receiving unsolicited support, like the following individual:

> Interviewer: So, have the people in your inner circle ever behaved in this kind of way, where they try to support you without you asking them to?
>
> Participant: Yeah, I would say so. But only if I'm doing something that I shouldn't be doing, or like you know, "That's good that you're trying to exercise right now." "That's good that you're trying to eat something healthy." (Black male, 43 years)

Interestingly, the extent to which support was solicited or unsolicited *intersected* with type of support. In the example above, the participant seemed to be describing unsolicited emotional support—providing compliments and encouragement without being asked to. In general, unsolicited emotional support generally seemed to be perceived positively by participants.

Other participants described receiving unsolicited instrumental support from their family members. Here, participants' reactions varied—some were positive, some were negative, and some expressed ambivalence, an attitude that indicated mixed feelings about the instrumental support they were receiving. One participant discussed an example of unsolicited instrumental support in which his wife would occasionally cook something healthier for dinner without telling him first. When asked whether he thought this was generally helpful or unhelpful, he seemed to be ambivalent:

> "I can't say either one. Really, it's not really helpful. I mean, I'm glad to have tried a lot of things and it just opens up more of my, you know, appreciation of the different types of foods that you can consume." (White male, 68 years)

The most common type of unsolicited support discussed by participants appeared to be informational support. Similar to unsolicited instrumental support, participants' reactions varied: some participants thought receiving this kind of support was helpful, but others did not. In one instance, a participant seemed to find unsolicited information about diabetes helpful, saying:

"Oh, I think it's great. I mean, I welcome it. You know, I openly accept it. I don't have any problems at all when someone, you know, gives me that kind of information." (Black male, 66 years)

Overall, unsolicited emotional support was generally perceived positively, but the extent to which instrumental or informational support was solicited or unsolicited did not seem to play a major role in participants' feelings about the support. Some participants seemed to welcome the unsolicited instrumental or informational support, whereas others did not. Other support dimensions that emerged from this study, such as directiveness (the extent to which a support partner was telling the participant what to do versus offering a suggestion), appeared to have more influence on participants' feelings about the instrumental and informational support they received from others and are discussed in the following sections.

### Theme 2: When it came to overprotective support, participants from both racial groups generally found close others doing things for them to be helpful, whereas participants' feelings about restrictions varied by race

Another support dimension of primary interest was overprotection, which has previously been defined in the literature as "unnecessary help", "excessive protection", or "attempts to restrict activities as a consequence of underestimating the patient's capabilities" [28]. We used this definition to ask participants specifically about instances in which they received support in which close others tried to do things for them that the participant could do themselves or restrict the participant's ability to do certain things. Similar to unsolicited support, overprotection seemed to intersect with type of support (e.g., instrumental support). For example, many of the participants described family members doing things for them that they could do themselves like getting groceries or cooking food. The following participant talked about preparing medication:

"Well, again, got to by starting with [my wife] because she's one who usually prepares the meds, she does that. Obviously, I can do it myself. But no, we kind of, you know, designated certain things that each of us do. And that's one of the things that she does on a regular basis." (Black male, 66 years)

When participants described behaviors that seemed to relate to tasks that the participant could do themselves, they did not seem to describe them as being "unnecessary" or providing "excessive protection"—rather, participants seemed to think these behaviors were quite helpful or desirable.

On the other hand, when participants responded to the part of the question that asked about behaviors in which close others restricted their ability to do certain things, almost all participants said that close others never restricted them. However, the way participants defined a "restriction" varied from person to person. Some participants drew a line by defining restrictions as someone directly commanding them to do something, as opposed to saying what they should or shouldn't do:

"Occasionally, they'll remind me that, 'We know you really like the cinnamon rolls, you know? But you really shouldn't have any.' 'Yeah, you're right.'. . .And it hasn't become a control issue, you know, where they're saying, 'Well, no, you definitely can't do that. Or you must do this now,' or anything like that." (White male, 60 years)

Other participants seemed to define a "restriction" as someone doing something tangible, as opposed to merely saying what they should or shouldn't do:

"No, nobody really restricts me. . .I'm pretty much, you know, I pay for my own groceries. So, nobody really restricts that. I mean, they may say, 'You probably shouldn't have bought that because it has blah, blah, blah, sugar in it', or this or that, or 'it has too much sodium' or something like that, but nobody really restricts." (Black female, 39 years)

Interestingly, even though this participant defined restrictions as a tangible act (e.g., restricting the foods that she buys or eats), she seemed reluctant to label behaviors as restrictions even when they were tangible. In the following example, this participant discussed how her close friend would buy diet beverages for her even if she asked for a non-diet beverage, or saw a candy the participant liked at the grocery store but specifically avoided getting it for her. When asked if she thought these were examples of her friend restricting her ability to do certain things, she responded:

"I wouldn't necessarily call it restricting. I feel like it's probably more of a concern. Like she's concerned about me and my diabetes. And so, she buys things that she knows are better suited for me. I don't really feel like it's a restriction. I don't view it that way." (Black female, 39 years)

Because the majority of participants stated that their close others never restricted their ability to do certain things, interviewers asked participants how they would feel if their close others did try to restrict them. Here, there appeared to be a race difference in participant responses. For many White participants, the idea of restrictions elicited a strong negative reaction:

"Well, if they tried to restrict my ability, that would be very, extremely annoying. Because I would not want to be restricted in the guise of help, like if somebody's going to go to the store and they're going to bring me some things. . .I guess if it was things that I wanted and needed, that would be fine. But if they're, again, restricting what I can have, because last time I checked, this was a free country." (White female, 60 years)

Among Black participants, responses were more mixed. Some Black participants reacted negatively:

"I think I would be insulted because I'm a grown ass man. I don't need to be coddled like a child. That's the way I would feel. And I'd be angered if someone would try to treat me like that." (Black male, 43 years)

However, some Black participants were more open to the idea of restrictions. As will be discussed more in Theme 4, some Black participants saw this kind of support as a sign of love:

"Well, I don't mind the help. . .Like I said before, that's concern and love. If you don't love nobody, you don't give a fuck what they do. Know what I mean? If you love somebody, you (inaudible). You talk to them, you harass them, you know what I mean? You don't harass them about doing shit, you harass them about things to help. I don't care if you harass me all day long. If you love me (inaudible) that means you love me." (Black male, 60 years)

In sum, when we defined overprotective support for participants during the interviews, they frequently discussed receiving support in which close others did things for them that they could do themselves and did not seem to find them excessive or unnecessary. However, White participants appeared to have a strong negative reaction to the idea of restrictions, whereas some Black participants reacted negatively but some saw it as a sign of love. It is worth noting that there were discrepancies in how participants defined restrictions, which seems to suggest that what is considered restrictive varies from person to person.

### Theme 3: In terms of support directiveness, participants from both racial groups generally expressed negative feelings when close others told them what to do, even if they found it helpful

As alluded to in the previous theme, some participants (both Black and White) believed there was an important distinction between being told what to do versus what was perceived more as a suggestion. This led us to believe that telling vs. suggesting, which we termed "directiveness," might be another intersecting dimension that has rarely been examined within the support literature but may be somewhat reflected in the social control literature [40, 41]. One participant clarified the difference between telling and suggesting:

"So, to me, yeah, to me, giving you a recipe isn't really advice, you know what I mean? Advice is more like, 'Well, you really shouldn't do this,' or 'You should do this more,' you know what I mean? But like, the suggestions are different than unsolicited advice, you know. So, if somebody gives me a suggestion, it's definitely more well-received than—it's like I'm five years old and you're my parent scolding me." (White male, 68 years)

When close others were suggesting rather than telling, sometimes participants added hedging language like "maybe" or framed their concerns as a question rather than a command. For example, a participant mentioned the following exchange with a close friend:

Participant: Beth? She's more gentle about stuff. She'll be like, "Oh, you're eating a Reuben for lunch?" I'm like, "Yeah, I'm only going to eat half of it though." "Oh okay." (laughs). So yeah, and—

Interviewer: So, I can imagine what she might have meant by that. But what do you think she was trying to say with that?

Participant: That it's okay to eat some carbs, but don't eat all of those things, you know. Don't eat the sandwich because there's dressing on it. (White female, 68 years)

Though several participants specifically expressed that they would rather their close others make suggestions instead of telling them what to do, telling was sometimes still seen as helpful. One Black participant discussed how he could feel both angry and grateful for the way his family members and Narcotics Anonymous sponsor support him:

"He's my NA sponsor, so he calls me out on it—and he's a diabetic, so we talk about diabetes a lot. He says, 'You know you got to do better than that, man. Don't let your diabetes get the best of you.' . . .Sometimes I might get mad about it, or feel some type of way. But after it's all said and done, when I have time to think about it and feed on it, I'm grateful, you know." (Black male, 65 years)

Another White participant also seemed to have mixed feelings towards being told what to do. Earlier in the interview, he discussed how his wife had "insisted" that he go to a class with a dietician, and that she would tell him not to eat certain things. When asked how it made him feel, he seemed to have negative feelings towards being told what to do, but at the same time, he also found her help useful:

"Sometimes I didn't like it. But I listened to her. I still have my feet, I still have my hands. (laughs) I can still see." (White male, 71 years)

On the whole, participants did not seem to like their close others telling them what to do. When participants expressed ambivalence, they seemed to feel displeasure with this kind of support but also found it to be effective in encouraging them to take care of diabetes.

### Theme 4: If support was perceived as inhibiting autonomy, it was generally undesired—but for Black participants, knowing that the support was provided out of love and care could make it more acceptable

Reactions to each of the different support dimensions, including aspects of unsolicited support and overprotective support, appeared to vary widely from person to person. However, for White participants, one of the major deciding factors for whether a support interaction was perceived positively or negatively seemed to depend on whether the participant felt like it was inhibiting their autonomy. This is distinct from the traditional definition of overprotective-ness, which names specific behaviors (e.g., doing things for an individual, restricting an individual) that are *theorized* to threaten autonomy. In these interviews, there was not always a specific support dimension or behavior that was perceived as a threat.

For one participant, a combination of being told what to do and being restricted appeared to serve as a threat to his autonomy:

"Well, if they tried to come in and say, 'Okay, we're going to do your shopping for you and only get you things that we deem are good for you,' I would be very resentful of that because that'd make me feel that they have decided that I'm incapable of dealing with my diabetes on my own. And I'd probably be very rude in telling them to go mind their own business." (White male, 60 years)

Another participant seemed to be open to close others doing things for her as long as it was delivered in a suggesting way:

"I would feel they overstepped. Probably if, you know, it's one thing to say, 'Can I help you?' or stop and give me something. Or maybe say, 'I saw this at the store, I thought maybe you'd like to try it?' That's one thing. But to just go ahead and do it. I don't know if I'd like that. I'm pretty independent." (White female, 54 years)

For this particular participant, it seemed that if her close others bought something for her but framed it as a suggestion, this would allow her the freedom to choose whether to try their suggestions or not. Thus, this participant's desire for maintaining her own autonomy seemed to connect to her desire for support that was more suggesting in nature.

For White participants, even if they knew the support was coming from a place of love or concern, the support was still undesired if it conflicted with their personal interests. One

participant reflected on whether her close friend telling her to eat healthier was a good thing or a bad thing:

> "I know she's only doing it out of a position of love. And it is a good thing. I just don't want to hear it. So, yeah. Am I going to eat cauliflower pizza? Hell no." (White female, 60 years)

Some Black participants also seemed to dislike support that they saw as inhibiting their autonomy. In response to a question about how she would feel if she received overprotective support from her close others, one participant remarked:

> "I would try to make an allowance for the fact that they care about me, but it would annoy me. It really would. Because I want to be in control of my own life, good or bad. You know, that's my business. If I crash and burn, I need to deal with the consequences. But I don't want to be handled like a child. You know, I'm an adult, you know? And I don't want anybody making decisions for me." (Black female, 50 years)

However, for some Black participants, supportive interactions seemed to be viewed positively if the participant saw love and care as one of the reasons for the support, even if the support appeared to inhibit their autonomy. When asked what his inner circle does to make him feel that they were "very supportive", one Black participant mentioned several examples of support from his family that involved telling him what to do:

> "They tell me when I was eating, 'Dad, quit drinking that beer. You need to quit drinking beer.' Or my fiancé: 'You can't eat all of that butter!' I mean, things they tell me about like a little kid. Which is good because it shows they care. If they didn't care, they wouldn't say nothing." (Black male, 60 years)

When the interviewer probed further about the participants' perspective that his family treated him like "a little kid," the participant indicated that these behaviors were desired because they showed that his family loved him:

> "Let me say this to you. When somebody's getting on you, right, about something you doing wrong, I'll give you some advice. That's love. That's all it is. If you're getting on me like that, it's love. So, I don't have a problem with that, it's love." (Black male, 60 years)

Some Black participants indicated an ambivalent attitude towards certain support interactions. One participant mentioned that her best friend was "on [her] constantly" about her blood glucose readings (Black, female, 39 years). When asked how she felt about this kind of support, the participant became emotional while discussing it:

> "I mean, it's necessary, because if somebody doesn't, I'm not going to either. Really, honestly. Sometimes I just wish she would just forget about it. But if nobody said anything to me, I would never keep it in the back of my mind. So sometimes you resent it, but then it's for the best because at least someone's concerned about how you're doing or where your health is taking you." (Black female, 39 years)

Her response seemed to indicate appreciation for the fact that these behaviors were done out of concern for the participant, yet the interactions left her unhappy. It could be that conflicts between a desire for autonomy and the knowledge that support was provided out of love

or care contributed to feelings of ambivalence towards certain support interactions among some Black participants.

To summarize, there were a variety of support behaviors that participants found undesirable—some did not like restrictions, whereas others did not like being told what to do. However, the common element that determined whether White participants deemed a given support behavior undesirable was whether it was perceived to inhibit their autonomy. Black participants also expressed some aversion to support that inhibited their autonomy; however, it seemed that the extent to which behaviors were perceived as being done out of love or care could counter the potential threat to their autonomy and make these behaviors more acceptable.

### Theme 5: Participants from both racial groups seemed to describe the tone that close others used to provide support, and this tone allowed participants to infer the reasons for the support being provided and may impact feelings towards support

During the coding process, we noticed that participants seemed to be trying to convey the tone their close others used during support interactions through their choice of words. In many cases, the close other's tone appeared to imply a subtext to the participant for why the support was being provided. Thus, tone appeared to be another important support dimension that has rarely been examined in the literature. Where possible, coders attempted to infer the different tones of support being conveyed through participants' descriptions of the support they received.

Some support behaviors were delivered in a way seemed to indicate they were evaluating what the participant should be doing to care for diabetes—thus, this support often appeared to be delivered with an *evaluative* tone. One participant recounted what her family members said to her during a family function, in which they seemed to use an evaluative tone:

> "They think, 'You shouldn't be eating that! It's sweet!' Or… 'You put all that sugar in your coffee?' 'No, because I don't use sugar!'" (Black female, 61 years)

This example appeared to be a more overt display of evaluative tone, in which the participant's family members made clear statements regarding what they believed the participant should or should not be doing to manage diabetes. However, implications of evaluation or criticism could also be more subtle. Another participant discussed something her husband might say when grocery shopping with him while using an evaluative tone:

> "I know that Jim looks at labels. So, it's like, 'Oh, this one has fifty-five carbs. You sure you want to buy that?'" (White female, 68 years)

In this example, the participant's husband was asking a question implying that he did not think the participant should buy that particular item without making an explicit statement.

Close others could imply other meanings with their tone as well. In a different instance, this participant seemed to infer that her mother was concerned about her through the way she asked a question, thereby showing that she was using a *concerned* tone:

> "Ever so often, she'll remind me, 'You watching your sugar?' Just usually jokingly, not as a serious type, you know, being annoyed or angry or anything like that. She's just always concerned. (White female, 60 years)

It is worth noting that the participant's mother did not explicitly state that they were concerned about the participant and their diabetes, at least based on the participant's recollection. Rather, participants seemed to use the tone with which the support was delivered as a way to infer that the reason their close others were providing support was out of worry or concern.

Another implied meaning that coders inferred from participant descriptions of a person's tone was warmth or love, which often arose when participants discussed how they were supported emotionally. One participant mentioned how his wife would provide him with compliments using a *warm* tone:

"Jill compliments me sometimes. She'll say, 'Hey, I can't believe that all the time you've had this and you're doing so well.' (White male, 71 years)

In another example, a participant discussed the way her older family members would tell her what to do in a warm, almost playful tone:

"They're not as direct. They'll just like—'Hey, baby. Hey, sweetie pie. How's it going? Are you keeping up with your weight loss? I know you are.' Like, they will answer before I answer. Like, it doesn't matter even if it is—'How you—?' and 'You eating right? I know you're not, I know you're not eating. Well, you know you better go ahead and eat, baby girl.' (laughs)" (Black female, 52 years)

The warm tone being used here appeared to be distinct from a tone where the close other appeared to be expressing concern about the participant's diabetes, or a tone implying what the participant should or should not be doing.

Importantly, a close other's tone could express concern, evaluation, and warmth, all within the same support interaction. One participant recalled the sorts of things his children would say to him when he would tell him what his blood glucose readings were:

"They say, 'Evidently, you're not doing what you're supposed to do. You're not eating right, Dad. You're not taking care yourself, you're starting to get really, really lax with this thing and your life is a game. What's wrong, Dad? You know, we love you and we don't want nothing to happen to you. . .You've got your grandkids; you don't love them enough to try to do right?'" (Black male, 63 years)

As to whether participants' feelings were dependent on the tone of the support, only a few participants explicitly indicated that this was the case, including the following participant:

"You know, and I guess it depends too, on the way they say it. You know, like Evelyn will say to me, just the way she says my name: 'Cynthia.' And I'm like, 'I don't want to hear it, Evelyn.' You know, things like that. Like, I guess it's the way that it's said, the tone of voice, the context in the conversation of where you, you know, that kind of thing." (White female, 60 years)

It is possible that the tone of support plays an important role in participants' reactions towards support; however, given that this theme was discovered through the coding process, we were not able to ask all participants about their reactions towards the support provider's tone. Future studies should further explore the importance of close others' tone within the context of support interactions.

## Discussion

With this qualitative study, we sought to better understand the most salient dimensions of support that Black and White individuals with type 2 diabetes experience from their close others. The analysis produced insights about participant perceptions of intrusive forms of support such as unsolicited and overprotective support, identified other support dimensions that have not been considered at length by prior research (i.e., directiveness and tone), and revealed specific factors (e.g., impact on autonomy) that may explain why certain support behaviors are seen as more desirable than others.

Across both Black and White participants, it was common for participants to discuss receiving unsolicited support of all types, including instrumental, emotional, and informational. Though some prior studies have found that unsolicited support was linked to negative outcomes [42, 43], these studies did not examine the effects of unsolicited support separately by type. In the current study, unsolicited emotional support was found to be desirable, whereas solicitation did not seem to affect whether instrumental or informational support was seen as desirable. Given our findings, it seems necessary to examine the effects of unsolicited support separately by type, as unsolicited emotional support may be more uniformly desirable and therefore have greater benefits for health outcomes than unsolicited instrumental and informational support.

In our exploration of overprotective support, we focused on two components: behaviors in which close others tried to do things for the participant that they could do themselves, and behaviors in which close others tried to restrict the participant's ability to do certain things. Most participants did not seem to object to their close others doing things for them—the more objectionable component of overprotective support for both Black and White individuals appeared to be restrictive behaviors. However, participants' definitions of what was considered restrictive varied considerably, from telling someone what to do to directly regulating what they could or could not do. There did not seem to be a clear consensus on what qualified as a restriction. Yet, White participants were strongly against the idea of restrictive behaviors, whereas some Black participants were open to restrictive behaviors. Given that this group of behaviors was not always perceived as overprotection, it seemed that "overprotective support" was not an apt label for these behaviors. It may be more appropriate to focus on the participants' *perceptions* of overprotectiveness, separate from the type of behaviors that are being engaged in (i.e., doing things for them or restricting them).

Together, our findings for unsolicited and overprotective support indicate an important consideration for future studies where unsolicited support and overprotective support are being measured quantitatively. "Intrusiveness," "negative interactions," and "unsupportive behaviors" may not be completely accurate descriptors of unsolicited support and overprotective support. These appear to be labels applied by researchers, not necessarily by participants themselves, especially given that not all forms of unsolicited and overprotective support were viewed as undesirable. Researchers should instead separate the measurement of the various dimensions of the support being received from the measurement of whether participants perceive these behaviors as helpful or harmful. Further, it will be crucial for researchers to directly examine whether unsolicited and overprotective support are helpful or harmful for health outcomes across a diverse set of racial/ethnic groups.

Our analysis also identified two additional support dimensions that are rarely considered in social support research—support directiveness and support tone. Support directiveness appeared to play an important role in determining the desirability of unsolicited instrumental or informational support. Several participants made distinctions between being told what to do versus being offered a suggestion, often preferring to receive suggestions from close others.

Thus, it might be more informative to focus on the directiveness of support rather than the solicitation of support as a determinant of outcomes.

The tone used by close others when delivering support seemed to provide a way for participants to infer the support provider's intentions without the support provider stating their intentions explicitly. It is possible that support tone plays an important role in determining whether a support behavior is desired or undesired. We were not able to fully examine this issue within the current study because this dimension of support was not anticipated; thus, the topic would greatly benefit from further exploration. Without interviewing participants' inner circle members, it is unclear whether participants were able to accurately infer a support partner's reasons behind providing support, which could lead to potential miscommunications. Future work should examine the extent to which the inferred reasons behind support and whether gaps between the inferred reason and the actual reason for support could be detrimental for health outcomes.

Of all the different factors discussed with participants, the most important factor in determining participants' responses towards support seemed to be whether the support was perceived to inhibit autonomy. If the support behavior was perceived to inhibit autonomy, the support was viewed negatively by both Black and White participants—regardless of the support dimension. However, for Black participants, if they recognized that the support was being provided because their close others loved and cared about them, this could make the support seem more acceptable and desirable, perhaps even counteracting the threat to their autonomy. Interestingly, this seems to align with a prior developmental study finding that showed intrusiveness was linked to beneficial social and emotional outcomes for Black children if maternal intrusiveness was experienced within the context of maternal warmth (e.g., verbal expressions of love) [17]. Future research should continue to explore whether perceptions of love and care might mitigate potential detrimental effects of support behaviors that appear to threaten autonomy for Black individuals with type 2 diabetes.

## Limitations

Several limitations to the current study are worth noting. First, the characteristics of our sample may affect the transferability of the findings, or the extent to which researchers can find these results applicable to their own particular research context [44]. Importantly, White participants appeared to be older than Black participants, and there was a greater percentage of White participants with a high household income or high level of education than Black participants. Future qualitative studies should engage in purposive sampling that ensures a more representative spread of age, income, and education so that participant experiences from each racial group are adequately represented. Additionally, the interviews occurred during the COVID-19 pandemic, which could be considered both a strength and a limitation. Participants' discussions of support interactions may provide useful insights into how individuals perceived their support interactions while in the midst of the COVID-19 pandemic, but these support interactions may be qualitatively different from support received prior to the start of the pandemic. Lastly, it would have been interesting to examine differences in perceptions of support dimensions from a more intersectional lens—that is, looking not just at differences by race, but race at the intersection of other identity categories such as gender, class, and sexual identity. Taking gender as an example, studies have shown that women are more likely to seek support than men [45], which could make women more likely to receive support as well. However, much of this initial research was conducted with White participants—more recent research has shown that Black couples are more egalitarian in their distribution of household activities than White couples [46], which may suggest more egalitarian support provision. We

were unable to conduct this kind of analysis at present given our sample size, but we urge future researchers to integrate intersectionality when attempting to understand group differences in social support interactions.

## Future directions

Stemming from this work, there are two lines of research that we believe would be particularly fruitful to pursue. First, researchers should conduct further qualitative studies on social support with a broader range of racial/ethnic minority groups. For example, a sizable literature has examined social support behaviors specifically among East Asian Americans and Latinx Americans [47], but these studies often use measures that were developed based on White American samples. Thus, we believe it would be beneficial for researchers to conduct inductive, qualitative studies to examine whether there are other dimensions of support that are relevant to other racial/ethnic groups, which would then inform the development of more culturally sensitive measures of social support. Second, the current study provides the impetus for future researchers to develop new quantitative measures of these more novel support dimensions, to examine race differences in the levels of these support dimensions, and to explore links between these support dimensions and health outcomes. For example, quantitative studies would be able to determine whether receiving more directive support or receiving support with a concerned tone is more common in one racial/ethnic group versus another or is more strongly linked to health for one racial/ethnic group versus another. As a whole, conducting these future studies will help ensure that our theoretical understanding of how support impacts health is grounded in the experiences of a diverse set of racial/ethnic groups. Once it is more clear which support dimensions are most beneficial for health, then we will be able to develop culturally sensitive interventions that more effectively utilize the support that close others provide.

## Conclusion

Intrusive forms of support like unsolicited support and overprotective support are often identified in the literature as negative or unsupportive, separate from the "good" kinds of support like instrumental, informational, and emotional support. The current study suggests that the solicitation and perceived overprotectiveness of support are just a few of many ways to describe support behaviors, and intersect with whether the support is instrumental, informational, or emotional. In general, this research contributes to a growing literature that examines race differences in social support more thoroughly. It was revealed that there are more dimensions that are salient to both Black and White participants than are being captured by the current measures available to us. Additionally, the extent to which support is perceived as inhibiting autonomy versus being provided out of love may shape reactions towards support more than the dimensions of support themselves, and may also vary by racial background. Overall, this research provides an initial step towards examining social support from a more culturally sensitive lens. Given how critical social support is for health, gaining a more culturally sensitive understanding of social support will help us identify the dimensions of support that are most beneficial while taking into account race and cultural context, thus helping to ensure better health for all.

## Supporting information

**S1 Fig. Recruitment diagram.**
(TIF)

**S1 File. Interview questions.**
(PDF)

**S2 File. Coding system.**
(PDF)

# Acknowledgments

The authors are indebted to Tiona Jones, Jennifer Melnyk, and Abigail Vaughn for assistance with recruitment; to Melissa Zajdel, Fiona Horner, Karen Lincoln, Aidan Wright, Kasey Creswell, Sheldon Cohen, Brooke Feeney, and Michael Trujillo for feedback on the study design; and to the participants who gave their time to this study.

# Author Contributions

**Conceptualization:** Jeanean B. Naqvi, Vicki S. Helgeson, Megan E. Hamm.

**Data curation:** Jeanean B. Naqvi, Rachael S. Liu.

**Formal analysis:** Jeanean B. Naqvi, Rachael S. Liu, Vicki S. Helgeson, Megan E. Hamm.

**Funding acquisition:** Jeanean B. Naqvi.

**Investigation:** Jeanean B. Naqvi, Rachael S. Liu.

**Methodology:** Jeanean B. Naqvi, Rachael S. Liu, Megan E. Hamm.

**Project administration:** Jeanean B. Naqvi, Rachael S. Liu.

**Supervision:** Vicki S. Helgeson.

**Writing – original draft:** Jeanean B. Naqvi.

**Writing – review & editing:** Jeanean B. Naqvi, Rachael S. Liu, Vicki S. Helgeson, Megan E. Hamm.

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
