## [Decision Letter · Decision Letter 0]

3 Mar 2023

PONE-D-23-00304Intrusive social support among Black and White individuals with type 2 diabetes: A “control issue” or a sign of “concern and love”?PLOS ONE

Dear Dr. Naqvi,

Thank you for submitting your manuscript to PLOS ONE. After careful consideration, we feel that it has merit but does not fully meet PLOS ONE’s publication criteria as it currently stands. Therefore, we invite you to submit a revised version of the manuscript that addresses the points raised during the review process.

I would like to sincerely apologise for the delay you have incurred with your submission. It has been exceptionally difficult to secure reviewers to evaluate your study. We have now received two completed reviews; the comments are available below. Please revise the manuscript to address all the reviewer's comments in a point-by-point response in order to ensure it is meeting the journal's publication criteria. Please note that the revised manuscript will need to undergo further review, we thus cannot at this point anticipate the outcome of the evaluation process.

We look forward to receiving your revised manuscript.

Kind regards,

Miquel Vall-llosera Camps

Senior Editor

PLOS ONE

Journal Requirements:

"This research was supported by National Institutes of Health F31 MD015922 and the Society for Personality and Social Psychology through the Jenessa Shapiro Graduate Research Award. The authors are indebted to Tiona Jones, Jennifer Melnyk, and Abigail Vaughn for assistance with recruitment; to Melissa Zajdel, Fiona Horner, Karen Lincoln, Aidan Wright, Kasey Creswell, Sheldon Cohen, Brooke Feeney, and Michael Trujillo for feedback on the study design; and to the participants who gave their time to this study."

"JN received funding from the National Institute on Minority Health and Health Disparities (https://www.nimhd.nih.gov/) F31MD015922 and the Society for Personality and Social Psychology (https://spsp.org/) Jenessa Shapiro Graduate Research Award. The funders had no role in study design, data collection and analysis, decision to publish, or preparation of the manuscript."

4. We noted in your submission details that a portion of your manuscript may have been presented or published elsewhere. [DETAILS AS NEEDED] Please clarify whether this [conference proceeding or publication] was peer-reviewed and formally published. If this work was previously peer-reviewed and published, in the cover letter please provide the reason that this work does not constitute dual publication and should be included in the current manuscript.

Additional Editor Comments:

We would expect qualitative studies to include the following: 1) defined objectives or research questions; 2) description of the sampling strategy, including rationale for the recruitment method, participant inclusion/exclusion criteria and the number of participants recruited; 3) detailed reporting of the data collection procedures; 4) data analysis procedures described in sufficient detail to enable replication; 5) a discussion of potential sources of bias; and 6) a discussion of limitations.

Reviewers' comments:

Reviewer's Responses to Questions

**Comments to the Author**

1. Is the manuscript technically sound, and do the data support the conclusions?

Reviewer #1: Yes

Reviewer #2: Yes

2. Has the statistical analysis been performed appropriately and rigorously? 

Reviewer #1: N/A

Reviewer #2: Yes

3. Have the authors made all data underlying the findings in their manuscript fully available?

Reviewer #1: Yes

Reviewer #2: Yes

4. Is the manuscript presented in an intelligible fashion and written in standard English?

Reviewer #1: Yes

Reviewer #2: Yes

5. Review Comments to the Author

Reviewer #1: As authors themselves have identified that this small sample size may not be representative of group. Use of black and while may be avoided and better use words like race/ Africans origin/ south American origin/ north American origin/ Hispanic/ Negros/ Indiana/ in place of black and white. The analysis / inference may be rewritten based on corrections to avoid language sounding racism . A simple method to assess and report thematic saturation in qualitative research

Greg Guest, plos paper should be read to report - specially point number 26 in table 1 showing how to summaries data-driven saturation studies may yield better presentation. abstract also need refining accordingly. oral testimony should have been grouped by making categories on the basis of responses, however that will require increasing the sample size.

Reviewer #2: Overall, this is a very well written manuscript and timely. Below are two very minor recommendations to enhance the manuscript for the readership:

Line 55, typically diabetes complications are described in addition to comborbidities. Consider rephrasing to diabetes related complications, and comorbid conditions such as cardiovascular disease and chronic kidney disease.

Please provide rationale for using Kahn and Antonucci’s convoy model. The current description is very helpful however additional information is warranted. Specifically, it will be helpful for readers to know a brief background of this model, what is the purpose of the model, and is it standard use in qualitative methods?

6. PLOS authors have the option to publish the peer review history of their article (what does this mean?). If published, this will include your full peer review and any attached files.

Reviewer #1: **Yes: **Neeta Kumar

Reviewer #2: No

---

## [Author Response · Author response to Decision Letter 0]

29 Mar 2023

Thank you for your feedback. Please see the attached document for a detailed response to comments from the editor and reviewers.

---

## [Decision Letter · Decision Letter 1]

20 Jun 2023

PONE-D-23-00304R1Intrusive social support among Black and White individuals with type 2 diabetes: A “control issue” or a sign of “concern and love”?PLOS ONE

Dear Dr. Naqvi,

Thank you for submitting your manuscript to PLOS ONE. After careful consideration, we feel that it has merit but does not fully meet PLOS ONE’s publication criteria as it currently stands. Therefore, we invite you to submit a revised version of the manuscript that addresses the points raised during the review process.

Thank you for your revisions:  please address the following:

Please change the consent to written consent.

Also I cannot see whether you have used COREQ checklist. Please can you let me know whether you have submitted this and if not please submit the COREQ checklist.

We look forward to receiving your revised manuscript.

Kind regards,

Julia Morgan

Academic Editor

PLOS ONE

Journal Requirements:

Reviewers' comments:

Reviewer's Responses to Questions

**Comments to the Author**

1. If the authors have adequately addressed your comments raised in a previous round of review and you feel that this manuscript is now acceptable for publication, you may indicate that here to bypass the “Comments to the Author” section, enter your conflict of interest statement in the “Confidential to Editor” section, and submit your "Accept" recommendation.

Reviewer #1: (No Response)

2. Is the manuscript technically sound, and do the data support the conclusions?

Reviewer #1: No

3. Has the statistical analysis been performed appropriately and rigorously? 

Reviewer #1: No

4. Have the authors made all data underlying the findings in their manuscript fully available?

Reviewer #1: Yes

5. Is the manuscript presented in an intelligible fashion and written in standard English?

Reviewer #1: No

6. Review Comments to the Author

Reviewer #1: Title: Intrusive social support among Black and White individuals with Type 2 diabetes: A “control issue” or a sign of “concern and love”?

the whole article is focusing on establishing that the black community gives care out of love and rest gives care out of the feeling of control. And generalizing this point on the whole community may not be quite right. Many lacunae are already pointed out by the authors themselves- and despite so many serious concerns- authors themselves understand and identified- publishing a paper with so many lacunae in design outcome implication-and so many drawbacks- with no substantial recommendation may not be worth publishing.

7. PLOS authors have the option to publish the peer review history of their article (what does this mean?). If published, this will include your full peer review and any attached files.

Reviewer #1: No

---

## [Author Response · Author response to Decision Letter 1]

21 Jun 2023

We thank the Academic Editor, Dr. Morgan, for her time reviewing and providing feedback on our manuscript. We have revised our discussion of the consent process to specify that written consent was provided. We have also included the COREQ checklist and made revisions to the paper regarding items on the checklist that were previously omitted from the paper.

---

## [Editor Report · Decision Letter 2]

22 Jun 2023

Intrusive social support among Black and White individuals with type 2 diabetes: A “control issue” or a sign of “concern and love”?

PONE-D-23-00304R2

Dear Dr. Naqvi, 

We’re pleased to inform you that your manuscript has been judged scientifically suitable for publication and will be formally accepted for publication once it meets all outstanding technical requirements.

Kind regards,

Julia Morgan

Academic Editor

PLOS ONE
---

## [Editor Report · Acceptance letter]

28 Jul 2023

PONE-D-23-00304R2 

Intrusive Social Support Among Black and White Individuals with Type 2 Diabetes: A “Control Issue” or a Sign of “Concern and Love”? 

Dear Dr. Naqvi:

I'm pleased to inform you that your manuscript has been deemed suitable for publication in PLOS ONE. Congratulations! Your manuscript is now with our production department. 

Kind regards, 

on behalf of

Dr. Julia Morgan 

Academic Editor

PLOS ONE